# A STATISTICAL THEORY OF COLD POSTERIORS IN DEEP NEURAL NETWORKS

**Laurence Aitchison**
Department of Computer Science,
University of Bristol,
Bristol, UK, F94W 9Q
laurence.aitchison@bristol.ac.uk

## ABSTRACT

To get Bayesian neural networks to perform comparably to standard neural networks it is usually necessary to artificially reduce uncertainty using a "tempered" or "cold" posterior. This is extremely concerning: if the generative model is accurate, Bayesian inference/decision theory is optimal, and any artificial changes to the posterior should harm performance. While this suggests that the prior may be at fault, here we argue that in fact, BNNs for image classification use the wrong likelihood. In particular, standard image benchmark datasets such as CIFAR-10 are carefully curated. We develop a generative model describing curation which gives a principled Bayesian account of cold posteriors, because the likelihood under this new generative model closely matches the tempered likelihoods used in past work.

## 1 INTRODUCTION

Recent work has highlighted that Bayesian neural networks (BNNs) typically have better predictive performance when we "sharpen" the posterior (Wenzel et al., 2020). In stochastic gradient Langevin dynamics (SGLD) (Welling & Teh, 2011), this can be achieved by multiplying the log-posterior by $1/T$, where the "temperature", $T$ is smaller than $1$ (Wenzel et al., 2020). Broadly the same effect can be achieved in variational inference by "tempering", i.e. downweighting the KL term. As noted in Wenzel et al. (2020), this approach has been used in many recent papers to obtain good performance, albeit without always emphasising the importance of this factor (Zhang et al., 2017; Bae et al., 2018; Osawa et al., 2019; Ashukha et al., 2020).

These results are puzzling if we take the usual Bayesian viewpoint, which says that the Bayesian posterior, used with the right prior, and in combination with Bayes decision theory should give optimal performance (Jaynes, 2003). Thus, these results may suggest we are using the wrong prior. While new priors have been suggested (e.g. Ober & Aitchison, 2020), they give only minor improvements in performance — certainly nothing like enough to close the gap to carefully trained non-Bayesian networks. In contrast, tempered posteriors directly give performance comparable to a carefully trained finite network.

The failure to develop an effective prior suggests that we should consider alternative explanations for the effectiveness of tempering. Here, we consider the possibility that it is predominantly (but not entirely) the *likelihood*, and not the prior that is at fault. In particular, we note that standard image benchmark datasets such as ImageNet and CIFAR-10 are carefully curated, and that it is important to consider this curation as part of our generative model. We develop a simplified generative model describing dataset curation which assumes that a datapoint is included in the dataset only if there is unanimous agreement on the class amongst multiple labellers. This model naturally multiplies the effect of each datapoint, and hence gives posteriors that closely match tempered or cold posteriors. We show that toy data drawn from our generative model of curation can give rise to optimal temperatures being smaller than $1$. Our model predicts that cold posteriors will not be helpful when the original underlying labels from all labellers are available. While these are not available for standard datasets such as CIFAR-10, we found a good proxy: the CIFAR-10H dataset (Peterson et al., 2019), in which $\sim 50$ humans annotators labelled the CIFAR-10 test-set (we use these as our training set, and use the standard CIFAR-10 training set for test-data). As expected, we find strong cold-posterior effects

when using the original single-label, which are almost entirely eliminated when using the 50 labels from CIFAR-10H. In addition, curation implies that each label is almost certain to be correct, which is one way to understand the statistical patterns exploited by cold posteriors. As such, if we destroy this pattern by adding noise to the labels, the cold posterior effect should disappear. We confirmed that with increasing label noise, the cold posterior effect disappears and eventually reverses (giving better performance at temperatures close to 1).

## 2 BACKGROUND: COLD AND TEMPERED POSTERIORS

Tempered (e.g. Zhang et al., 2017) and cold (Wenzel et al., 2020) posteriors differ slightly in how they apply the temperature parameter. For cold posteriors, we scale the whole posterior, whereas tempering is a method typically applied in variational inference, and corresponds to scaling the likelihood but not the prior,

$$\log \mathrm{P}_{\text{cold}}\left(\theta|X,Y\right) = \tfrac{1}{T}\log \mathrm{P}\left(X,Y|\theta\right) + \tfrac{1}{T}\log \mathrm{P}\left(\theta\right) + \text{const} \tag{1}$$

$$\log \mathrm{P}_{\text{tempered}}\left(\theta|X,Y\right) = \tfrac{1}{\lambda}\log \mathrm{P}\left(X,Y|\theta\right) + \quad \log \mathrm{P}\left(\theta\right) + \text{const}. \tag{2}$$

While cold posteriors are typically used in SGLD, tempered posteriors are usually targeted by variational methods. In particular, variational methods apply temperature scaling to the KL-divergence between the approximate posterior, $\mathrm{Q}\left(\theta\right)$ and prior,

$$\mathcal{L} = \mathbb{E}_{\mathrm{Q}(\theta)}\left[\log \mathrm{P}\left(X,Y|\theta\right)\right] - \lambda\,\mathrm{D}_{\text{KL}}\left(\mathrm{Q}\left(\theta\right)||\,\mathrm{P}\left(\theta\right)\right). \tag{3}$$

Note that the only difference between cold and tempered posteriors is whether we scale the prior, and if we have Gaussian priors over the parameters (the usual case in Bayesian neural networks), this scaling can be absorbed into the prior variance,

$$\tfrac{1}{T}\log \mathrm{P}_{\text{cold}}\left(\theta\right) = -\tfrac{1}{2T\sigma_{\text{cold}}^2}\sum_i \theta_i^2 + \text{const} = -\tfrac{1}{2\sigma_{\text{tempered}}^2}\sum_i \theta_i^2 + \text{const} = \log \mathrm{P}_{\text{cold}}\left(\theta\right). \tag{4}$$

in which case, $\sigma_{\text{cold}}^2 = \sigma_{\text{tempered}}^2/T$, so the tempered posteriors we discuss are equivalent to cold posteriors with rescaled prior variances.

## 3 METHODS: A GENERATIVE MODEL FOR CURATED DATASETS

Standard image datasets such as CIFAR-10 and ImageNet are carefully curated to include only unambiguous examples of each class. For instance, in CIFAR-10, student labellers were paid per hour (rather than per image), were instructed that "It's worse to include one that shouldn't be included than to exclude one", and then Krizhevsky (2009) "personally verified every label submitted by the labellers". For ImageNet, Deng et al. (2009) required the consensus of a number of Amazon Mechanical Turk labellers before including an image in the dataset.

To understand the statistical patterns that might emerge in these curated datasets, we consider a highly simplified generative model of consensus-formation. In particular, we draw a random image $X$ from some underlying distribution over images, $\mathrm{P}\left(X\right)$, and ask $S$ humans to assign a label, $\{Y_s\}_{s=1}^S$ (e.g. using Mechanical Turk). We force every labeller to label every image and if the image is ambiguous they are instructed to give a random label. If all the labellers agree, $Y_1 = Y_2 = \cdots = Y_S$, consensus is reached and we include the datapoint in the dataset. If any of the labellers disagree consensus is not reached, and we exclude the datapoint (Fig. 1), Formally, the observed random variable, $Y$, is taken to be the usual label if consensus was reached and `None` if consensus was not reached (Fig. 2B),

$$Y|\{Y_s\}_{s=1}^S = \begin{cases} Y_1 & \text{if } Y_1 = Y_2 = \cdots = Y_S \\ \texttt{None} & \text{otherwise} \end{cases} \tag{5}$$

Taking the human labels, $Y_s$, to come from the set $\mathcal{Y}$, so $Y_s \in \mathcal{Y}$, the consensus label, $Y$, could be any of the underlying labels in $\mathcal{Y}$, or `None` if no consensus is reached, so $Y \in \mathcal{Y} \cup \{\texttt{None}\}$. When consensus was reached, the likelihood is,

$$\mathrm{P}\left(Y=y|X,\theta\right) = \mathrm{P}\left(\{Y_s=y\}_{s=1}^S|X,\theta\right) = \prod_{s=1}^S \mathrm{P}\left(Y_s=y|X,\theta\right) = \mathrm{P}\left(Y_s=y|X,\theta\right)^S \tag{6}$$

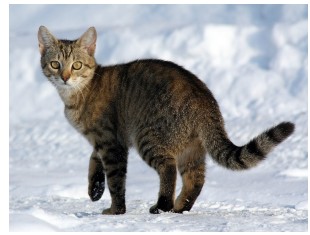 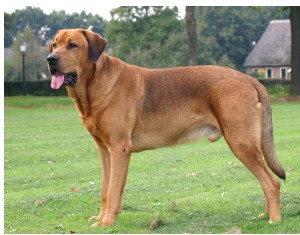 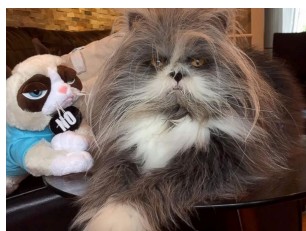

$Y_1 = Y_2 = Y_3 = \text{cat}$   $Y_1 = Y_2 = Y_3 = \text{dog}$   $Y_1 = Y_2 = \text{cat};$   $Y_3 = \text{dog}$

$Y = \text{cat}$   $Y = \text{dog}$   $Y = \text{None}$

Figure 1: A simple example of our generative process describing dataset curation, with $S = 3$. Labellers have the task of classifying images as cats or dogs. The first image is unambiguously a cat, so the three labellers agree, consensus is reached, and the image is included in the dataset. The second image is unambiguously a dog, the labellers agree, consensus is reached, and the image is included in the dataset. However, the third image is ambiguous: the labellers disagree, consensus is not reached, and the image may be excluded from the dataset.

where we have assumed labellers are IID. This would appear to directly give an account of tempering, as we have taken the single-labeller likelihood to the power $S$, which is equivalent to setting $\lambda = 1/S$. However, to see how the full generative model functions we need to go on to consider the case in which consensus was not reached,

$$\mathrm{P}\left(Y = \text{None}|X, \theta\right) = 1 - \sum_{y \in \mathcal{Y}} \mathrm{P}\left(Y = y|X, \theta\right) = 1 - \sum_{y \in \mathcal{Y}} \mathrm{P}\left(Y_s = y|X, \theta\right)^S. \tag{7}$$

To understand the impact of our model of consensus formation, note that the probability of a particular class-label can be separated into two terms, a probability of consensus, and the probability of a particular class given that consensus was reached,

$$\mathrm{P}\left(Y = y|X, \theta\right) = \mathrm{P}\left(Y = y|Y \neq \text{None}, X, \theta\right) \mathrm{P}\left(Y \neq \text{None}|X, \theta\right). \tag{8}$$

Remarkably, knowing there was consensus gives us information about the weights, even in the absence of the class label,

$$\mathrm{P}\left(Y \neq \text{None}|X, \theta\right) = \sum_{y \in \mathcal{Y}} \mathrm{P}\left(Y_s = y|X, \theta\right)^S, \tag{9}$$

in essence, telling us that the output probability was close to 1 for one of the class labels, without telling us which one. Schematically (Fig. 3), we see that the datapoints with a consensus class-label, "cat" or "dog", lie far from the decision boundary where the class is unambiguous, and consensus is easily reached. In contrast, in regions close to the decision boundary the inputs are ambiguous, which tends to produce disagreement in the labellers, leading to noconsensus. Thus, the existence of one or more consensus points in a region implies that decision boundaries do not go through that region, giving us information about the decision boundary location, even if the label is not known. Concurrent work has shown that this likelihood can be used to explain classical semi-supervised likelihoods (Aitchison, 2020), so this term really does give information about the neural network parameters. Finally, the label probability, conditioned on consensus,

$$\mathrm{P}\left(Y = y|Y \neq \text{None}, X, \theta\right) = \frac{\mathrm{P}\left(Y = y|X, \theta\right)}{\mathrm{P}\left(Y \neq \text{None}|X, \theta\right)} = \frac{\mathrm{P}\left(Y_s = y|X, \theta\right)^S}{\sum_{y \in \mathcal{Y}} \mathrm{P}\left(Y_s = y|X, \theta\right)^S} \tag{10}$$

simply represents a "reparameterised" softmax.

In datasets where the noconsensus inputs are known it is clear we should use the full likelihood, (Eq. 6 and 7). The question is: in real-world datasets, where we only know the consensus inputs and the noconsensus inputs have been thrown away (Fig. 2C), can we use Eq. (10), a reparameterisation of the softmax-categorical probabilities, for the known consensus points? The answer is no because in Bayesian inference, we do not get to just pick a sensible-looking conditional probability distribution, such as Eq. 10, to use as the likelihood. Instead, we need to write down the full generative model,

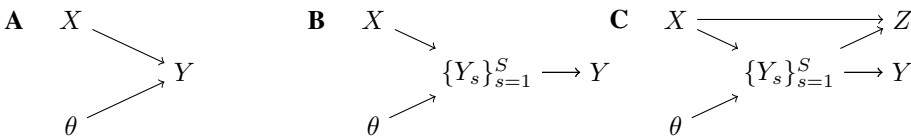

Figure 2: **A** The standard generative model for supervised tasks, assuming no data curation. **B** Generative model of data curation, where the consensus and noconsensus images are known. **C** Generative model of data curation, where the noconsensus images are unknown. The underlying images, $X$ are no longer observed. Instead, we observe $Z$, where $Z = X$ if consensus is reached ($Y_1 = Y_2 = \cdots = Y_S$) and $Z = $ None otherwise.

and marginalise over unknown latents. To write down the generative model in the case of unknown noconsensus images (Fig. 2C), we need to take $X$ to be an unobserved latent variable, and take $Z$ to be an observed random variable,

$$Z|X, \{Y_s\}_{s=1}^S = \begin{cases} X & \text{if } Y_1 = Y_2 = \cdots = Y_s \\ \text{None} & \text{otherwise} \end{cases} \tag{11}$$

which is the underlying image, $X$, if consensus is reached, and None otherwise. As $Z$ depends on $\theta$ (Fig. 2C), we cannot take the usual shortcut of using $\mathrm{P}\left(Y|Z, \theta\right)$; we must instead use the full likelihood, $\mathrm{P}\left(Y, Z|\theta\right)$,

$$\mathrm{P}\left(Y, Z|\theta\right) = \sum_{\{Y_s\}_{s=1}^S} \int dX\, \mathrm{P}\left(X, Y, \{Y_s\}_{s=1}^S, Z|\theta\right)$$

$$= \sum_{\{Y_s\}_{s=1}^S} \int dX\, \mathrm{P}\left(X\right) \left[\textstyle\prod_{s=1}^S \mathrm{P}\left(Y_s|X, \theta\right)\right] \mathrm{P}\left(Y|\{Y_s\}_{s=1}^S\right) \mathrm{P}\left(Z|X, \{Y_s\}_{s=1}^S\right). \tag{12}$$

for $y$ not None,

$$\mathrm{P}\left(Y = y|\{Y_s\}_{s=1}^S\right) = \begin{cases} 1 & \text{if } Y_s = y \text{ for all } s \in \{1, \ldots, S\} \\ 0 & \text{otherwise} \end{cases} \tag{13}$$

and,

$$\mathrm{P}\left(Z|X, \{Y_s\}_{s=1}^S\right) = \begin{cases} \delta(Z - X) & \text{if } Y_1 = Y_2 = \cdots = Y_S \\ \mathbb{I}_{Z=\text{None}} & \text{otherwise} \end{cases} \tag{14}$$

where $\delta$ is the Dirac-delta function, and where the indicator function, $\mathbb{I}_{Z=\text{None}}$ is 1 if $Z = $ None and 0 otherwise. Substituting Eq. (13) and Eq. (14) into Eq. (12),

$$\mathrm{P}\left(Y = y, Z|\theta\right) = \mathrm{P}\left(X\right) \prod_{s=1}^S \mathrm{P}\left(Y_s = y|X, \theta\right) \propto \mathrm{P}\left(Y_s = y|X, \theta\right)^S, \tag{15}$$

where the proportionality arises because $\mathrm{P}\left(X\right)$ does not depend on the parameters of interest, $\theta$. Note that this is proportional to Eq. (6) above, and not Eq. (10), so this does not just represent a reparameterisation of the softmax.

Finally, this explanation for the cold posterior effect would suggest that we would see cold posteriors even in maximum a-posteriori (MAP) inference, as we confirm in Appendix A. This is expected as it is "common knowledge" (but we do not know of a good reference) that while weight-decay is closely related to MAP inference with Gaussian priors, the best performing value of the weight decay coefficient tend to be lower than those suggested by untempered MAP inference.

## 4  DIFFERENCES BETWEEN COLD POSTERIOR AND DATASET CURATION SETUPS

Our model of data curation provides a direct explanation for the effectiveness of cold posteriors, as Eq. (6) takes the underlying likelihoods, $\mathrm{P}\left(Y_s|X, \theta\right)$ to the power $S$, which has exactly the same

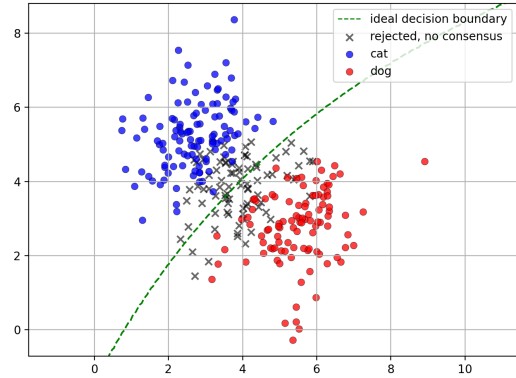

Figure 3: Illustrative example of artificial clustering induced in the dataset by rejection of ambiguous no-consensus points. The input points for "cat" and "dog" classes are generated from separate 2D Gaussian distributions, and the classifier (and decision boundary) comes from the ratio of the Gaussian probability density functions. For the consensus processes, we used $S = 7$ (we used a relatively large value to make the effects unambiguous).

effect as $1/\lambda$ in tempered posteriors (Eq. 2). However, it is important to note two differences between the cold posterior setup and the ideal setup for our generative model. First, our generative model assumes that the noconsensus points are known, or if they are unknown, we can compute the integral,

$$\mathrm{P}\left(Y\!=\!\texttt{None}, Z\!=\!\texttt{None}|\theta\right) = \int dX \ \mathrm{P}\left(X\right) \left(1 - \sum_{y \in \mathcal{Y}} \mathrm{P}\left(Y_s = y | X, \theta\right)^S\right) \tag{16}$$

which is obtained by evaluating Eq. (12) in the case where $Y\!=\!\texttt{None}$, and substituting Eq. (7). Of course, in reality it is not possible to compute this integral because we do not know the underlying $\mathrm{P}\left(X\right)$ (and usually, we do not even have samples from this distribution). In contrast, the standard cold-posterior setup entirely ignores these terms. Second, in the usual setting, the test data is also subject to the same consensus-formation process, in which case, we should use Eq. (10) for prediction. In contrast, in the standard cold-posterior setting, we use the single-labeller distribution, $\mathrm{P}\left(Y_s|X, \theta\right)$. (Note that if the test-set was drawn "from the wild", without dataset curation, and labelled by a single labeller, then we should use $\mathrm{P}\left(Y_s|X, \theta\right)$ for prediction). Overall then, while our model of data curation offers a potential explanation of the benefits of tempering, the differences in setup imply that we cannot expect $\lambda^* = 1/S$, to hold exactly, where $\lambda^*$ is the optimal temperature, and this is confirmed in our results on toy data below.

## 5 RESULTS

### 5.1 DATA SAMPLED FROM A KNOWN GAUSSIAN PROCESS MODEL

In the introduction, we took it as given that if the prior is correct, the optimal approach is to use the true Bayesian posterior, avoiding either tempering or cold posteriors. To check that this is indeed the case, we tested the performance, measured as test-log-likelihood for tempered posteriors using data generated from a known Gaussian process model. We uniformly generated 50 input points on the 1D interval [-10, 10], and used a Gaussian process with squared exponential kernel with a standard deviation of $4$, and kernel bandwidth of $1$. For inference, we used reparameterised VI, with an approximate posterior given by multiplying the prior by a single Gaussian factor for each datapoint (Ober & Aitchison, 2020).

In particular, we used a Gaussian process prior for the function values, $\mathbf{u}$, at the training inputs (Williams & Rasmussen, 2006),

$$\mathrm{P}\left(\mathbf{f}|\mathbf{x}\right) = \mathcal{N}\left(\mathbf{f}; \mathbf{0}, \mathbf{K}(\mathbf{x})\right) \tag{17}$$

and we use an approximate posterior (Hensman et al., 2015; Matthews et al., 2016; Ober & Aitchison, 2020) defined by multiplying the prior by a Gaussian with diagonal covariance, where we treat $\mathbf{v}$ and

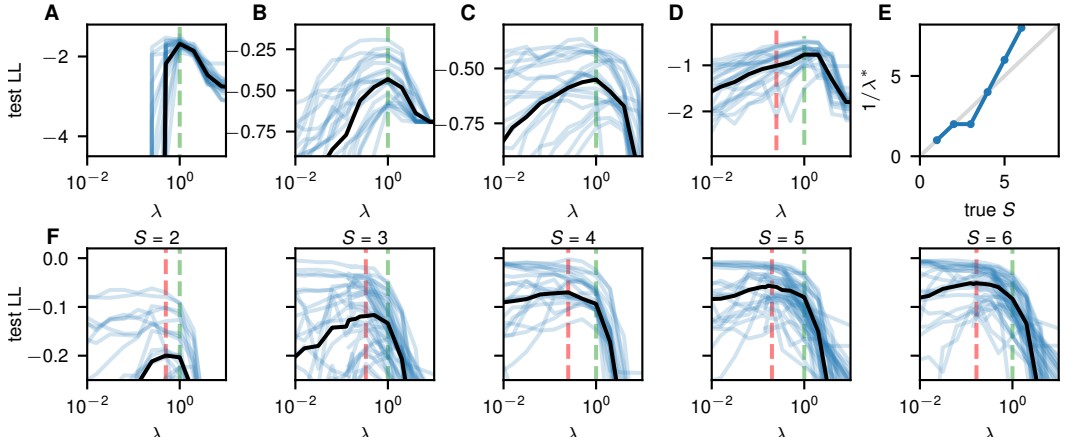

Figure 4: Tempering with data generated from a toy Gaussian process model. **A** Performance of tempered posteriors on GP regression. The green dashed line highlights $\lambda = 1$. The black line represents the mean of 20 runs (translucent blue lines). **B** GP classification. **C** GP classification with our model of consensus formation. We train and test using our exact log-likelihood, and we include knowledge of the no-consensus input datapoints in the datasets. **D** Data generated from our model of consensus formation, with $S = 4$. Training uses $\mathrm{P}\left(Y_s|X, \theta\right)$ log-likelihood (excluding noconsensus points), but testing uses the exact test-log-likelihood (Eq. 6 and 7). The red dashed line lies at $\lambda = 1/S = 1/4$. **E** Data generated from our model of consensus formation, with $S = 4$. Training and testing follow the standard cold-posterior setup, excluding noconsensus points and using $\mathrm{P}\left(Y_s|X, \theta\right)$. The x-axis gives the number of labellers in the underlying generative process, and the y-axis gives the reciprocal of the optimal temperature. **F** Plots underlying **E**. The red dashed lines indicate $\lambda = 1/S$.

$\mathbf{\Lambda}^{-1}$ as variational parameters.

$$\mathrm{Q}\left(\mathbf{f}\right) \propto \mathrm{P}\left(\mathbf{f}|\mathbf{x}\right) \mathcal{N}\left(\mathbf{v}; \mathbf{f}, \mathbf{\Lambda}^{-1}\right) = \mathcal{N}\left(\mathbf{f}; \mathbf{\Sigma}\mathbf{\Lambda}\mathbf{v}, \mathbf{\Sigma}\right) \quad \text{where} \quad \mathbf{\Sigma} = \left(\mathbf{K}^{-1}(\mathbf{x}) + \mathbf{\Lambda}\right)^{-1} \quad (18)$$

Note that this captures the true posterior in the case of regression (where $\mathbf{v}$ is set to $\mathbf{y}$ and $\mathbf{\Lambda}^{-1}$ is the output noise covariance; Ober & Aitchison, 2020). We then optimize the tempered ELBO (e.g. Zhang et al., 2017),

$$\mathcal{L} = \mathbb{E}_{\mathrm{Q}(\mathbf{f})}\left[\log \mathrm{P}\left(\mathbf{y}|\mathbf{f}\right) + \lambda\left(\log \mathrm{P}\left(\mathbf{f}|\mathbf{x}\right) - \log \mathrm{Q}\left(\mathbf{f}\right)\right)\right]. \quad (19)$$

And we used the standard GP approach for prediction at test points (Williams & Rasmussen, 2006).

Initially, we tried using standard regression and classification generative models, without including our model of consensus formation. Unsurprisingly, $\lambda = 1$, corresponding to the Bayesian posterior, is optimal for GP regression (Fig. 4A) and classification (Fig. 4B). For GP regression, we use a Gaussian likelihood with standard deviation 1, and for classification, we use a standard sigmoid probability with a Bernoulli likelihood.

Next, we confirmed that even under our new, more complex generative model of dataset curation $\lambda = 1$ was still optimal (Fig. 4C), if we trained and tested using the correct form for the log-likelihood. In particular, we considered no-consensus inputs as known and used Eq. 7 to incorporate their likelihood. However, in standard benchmarks, only the consensus inputs are known. As such, next we generated curated data, trained using $\mathrm{P}\left(Y_s|X, \theta\right)$ and ignoring noconsensus points, but tested on the correct likelihood, including noconsensus points (Fig. 4D); the optimal temperature remains around 1. Finally, we considered the standard cold/tempered posterior setup, where we use $\mathrm{P}\left(Y_s|X, \theta\right)$, excluding noconsensus points, for training and testing. The optimal $\lambda$ indeed fell with $S$ (Fig. 4EF), giving a potential explanation for the cold posterior effect. As expected, we have $S \approx 1/\lambda^*$, but the relation does not appear to hold exactly due to the mismatches discussed in Sec. 4. Finally, we show in Appendix B that the same patterns emerge in a Bayesian neural network, using Langevin sampling for inference.

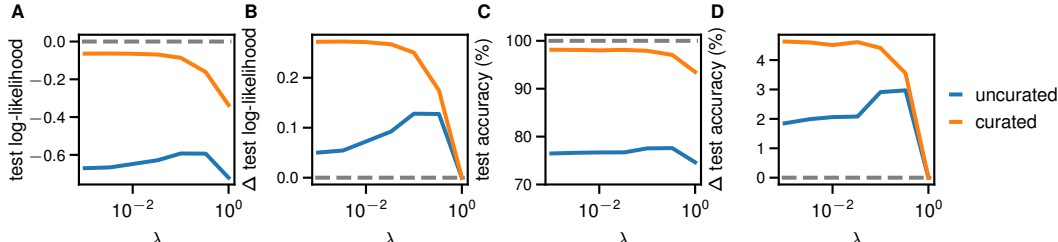

Figure 5: **A** Test log-likelihood with tempering. **B** Change in test log-likelihoods from the baseline at $\lambda = 1$. We see standard cold posterior effects at low noise levels, which reverse at higher noies levels. **C, D** As **A, B** but for test accuracy.

## 5.2 Curated and uncurated Galaxy Zoo datasets

The most direct test of our theory is to evaluate the cold-posterior effect on real-world curated and uncurated datasets. To this end, we used the Galazy Zoo 2 dataset (Willett et al., 2013), as the original dataset is not "curated" in our sense (the criteria for inclusion were e.g. brightness) and the dataset gives classifications from $\sim 50$ labellers. We used a reduced label set for simplicity (see Appendix C). For the uncurated dataset, we selected 12500 points at random from the full dataset, and for the curated dataset, we selected the 12500 most confident points (defined in terms of labeller agreement), such that the correct class-balance was maintained. These datasets were split at random into 2500 training points and 10000 test points. Note that using our generative model directly would lead to drastically different class-balance in the curated and uncurated datasets, due to different levels of certainty for different classes. To perform probabilistic inference, we used SGLD with code adapted from (Zhang et al., 2019). In particular, we used their standard settings of a ResNet18, momentum of 0.9, and a cyclic cosine learning rate schedule. Due to the smaller size of our training set, we used longer cycles (600 epochs rather than 50), and more cycles (8 rather than 4).

As expected, we found that performance on curated data was far better than that on uncurated data, both in terms of test log-likelihood and test accuracy (Fig. 5AC). We found some cold-posterior effects in uncurated data, which is not surprising because there may be other causes of cold-posteriors such as model-mismatch or biases in SGLD. Critically though, we found much stronger cold-posterior effects for curated data than for uncurated data (Fig. 5BD). Moreover, these plots tend to understate the differences between curated and uncurated data. In particular, the proportional changes to the test-log-likelihood (which has an upper bound of zero) and the test error is for curated data is very large, with both changing by a factor of $\sim 3$ as temperature falls. In contrast, proportional changes for test-log-likelihood and test-error are much smaller (only $\sim 20\%$).

## 5.3 CIFAR-10H

One prediction made by our framework is that if we had access to the original underlying human labels on the full dataset, including images that were rejected because consensus was not reached, then tempering should not be necessary. Obviously, all this additional information is not available for standard datasets such as CIFAR-10. However, we are able to get close by considering the CIFAR-10H dataset (Peterson et al., 2019). The authors of this work asked around 50 human labellers to label the CIFAR-10 test-set. As we might expect given the careful curation that went in to creating the original CIFAR-10 dataset (Krizhevsky, 2009), for almost half of the datapoints, all $\sim 50$ labellers agreed, and more than three-quarters of images had 2 or fewer disagreements (corresponding to 4%) of labellers (Fig. 6A). While it is not possible to estimate $S$ without having information about the unknown image distribution, the high level of agreement would indicate that the effective value of $S$ is large — potentially even larger than 10.

Next, we trained a neural network on the $\sim 50$ labels provided by the CIFAR-10H dataset. As these labels are provided for the test-set with 10,000 images, this required us to swap the identities of the test and training sets (so that our training set consisted of 10,000 points, each with around 50 labels, and the test set consists of 50,000 points, with the single label from the original CIFAR-10 dataset). We compared against training with the standard CIFAR-10 test set, containing the same images but

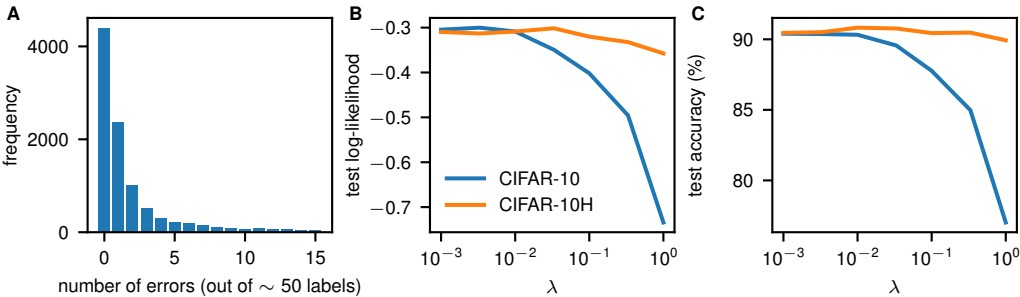

Figure 6: **A** The number of errors (defined as labellers who disagree with the most popular category) out of around 50 labels. **B** The test log-likelihood for different values of $\lambda$ for training with the standard single-labels provided by CIFAR-10 and the 50 labels provided by CIFAR-10H. Note that here we have swapped the identities of the train and test set. **C** As in **B**, but for accuracy.

only with a single label. To perform probabilistic inference, we used SGLD with code adapted from (Zhang et al., 2019). In particular, we used their standard settings of a ResNet18, momentum of 0.9, and a cyclic cosine learning rate schedule. Due to the smaller size of our training set, we used longer cycles (150 epochs rather than 50), and more cycles (8 rather than 4). Importantly, we kept all the parameters of the learning algorithm the same for CIFAR-10H and our CIFAR-10 comparison. The only complication was that to keep the same effective learning rate for CIFAR-10H we need to take into account the number of labellers per datapoint (if we have 50 labels for a datapoint, the loss and the gradients of the loss are 50 times larger, resulting in step-sizes that are also 50 times larger). As such, to keep the same step-sizes, we divided the actual learning rate by 50. An alternative way to look at this is that we use the average log-likelihood per labeller (rather than per-image). Importantly, this change in learning rate, leaves the stationary distribution was unchanged, as changing the learning rate is *not* equivalent to tempering. All other parameters were left at the values specified by Zhang et al. (2019).

As expected, when training on the single-label from the original CIFAR-10 testset (blue lines Fig. 6BC), there are very large tempering effects. In contrast, when training on the $\sim 50$ labels provided CIFAR-10H (orange), the effects of tempering are far smaller. In a sense, this is not surprising — using 50 labellers in effect makes the likelihood 50 times stronger, which is very similar to applying tempering. But this simplicity is the point: in the Bayesian setting we need to condition on all data — in our case all the labels, and once we do that, the cold posterior effect disappears.

Note that while tempering-effects are dramatically reduced, they have not been eliminated entirely. This is expected in our setting, both because of the mismatch between this setup and the exact setup (Sec. 4), but also because the inference method, SGLD, becomes more accurate as the minibatch-size increases, but we can never use full-batch in practical settings due to the large size of datasets such as CIFAR-10 (Welling & Teh, 2011). In SGLD minibatch gradients are used as a proxy for the gradient for the full dataset, but these minibatch gradients contain additional noise, and there is a potential that reducing the temperature may partially compensate for this additional noise. This is particularly evident if we consider (Wenzel et al., 2020, Fig. 6), which showed that cold-posterior effects can be amplified by using very small minibatch sizes — smaller minibatches imply larger variance in gradient estimates, and hence more potential for lower temperatures to compensate for that additional noise. That said, these effects do not appear to be significant for CIFAR-10 (Wenzel et al., 2020, Fig. 5), so we are in agreement with Wenzel et al. (2020) that minibatch noise are unlikely to be the primary source of tempering effects.

## 5.4 CIFAR-10 WITH NOISY LABELS

Curated labels are almost certain to represent the true class, and this is one way to understand the statistical patterns induced by curation that are exploited by tempered posteriors. We therefore considered disrupting this property by adding noise to the labels. In the standard classification setting, this should have very little effect (except to shrink the logit outputs somewhat to give more

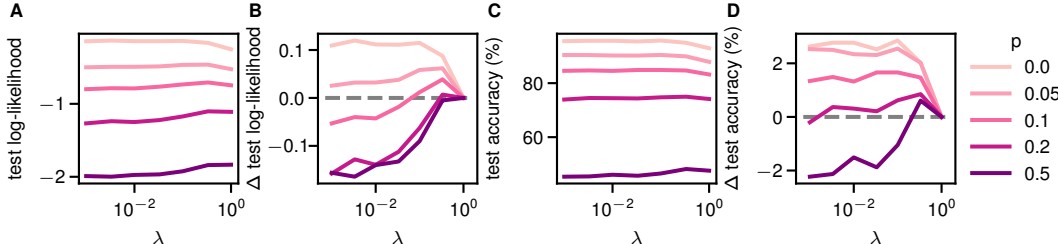

Figure 7: **A** Test log-likelihood with tempering for different probabilities of using a noisy label, $p$. Noiser labels mean test-log-likelihoods are lower. **B** Change in test log-likelihoods from the baseline at $\lambda = 1$. We see standard cold posterior effects at low noise levels, which reverse at higher noies levels. **C, D** As **A, B** but for test accuracy.

uncertain outputs). However, and as expected under our theory, we find that adding noise destroys and eventually reverses the cold-posterior effect (Fig. 7). This confirms that cold-posteriors are exploiting specific properties of the labels that are destroyed by adding noise. As such, cold posteriors are not likely to arise e.g. from failing to capture the prior over neural network weights (in which case adding noise to the outputs should have little effect). We suspect that the small improvement in performance at the first value of $\lambda$ below 1 might be due to partial compensation for additional noise introduced by minibatch estimates of the gradient.

## 6 RELATED WORK

Wenzel et al. (2020) introduced the cold-posterior effect in the context of neural networks, and proposed and dismissed multiple potential explanations (although none like the one we propose here).

Other past work, though not in the neural network context, argues that tempering may be important in the context of model misspecification (Grünwald, 2012; Grünwald et al., 2017). Critically, we believe that there may be many causes of cold-posterior like effects, including but not limited to curation, model misspecification and artifacts from SGLD. Ultimately, the contribution of each of these factors in any given setting will depend on the exact dataset and inference method in question. Importantly, this also means we do not necessarily expect there to be no tempering in uncurated data, merely that we should see *less* tempering in the case of uncurated data.

Finally, the closest paper is concurrent work (Adlam et al., 2020) raising the possibility that BNNs overestimate aleatoric uncertainty, in part because of high-quality labels available in benchmark datasets. However, they concluded that while cold posteriors might help us to capture our priors, they do not correspond to an exact inference procedure. In contrast, here we give a generative model of dataset curation in which tempered likelihoods emerge naturally even under exact inference methods.

## 7 DISCUSSION AND CONCLUSIONS

We showed that modelling the process of data-curation can explain the improved performance of tempering or cold posteriors in Bayesian neural networks. While this work does provide a justification for future practitioners to use tempering in their Bayesian neural networks, we would urge caution. Importantly, and as we confirmed in Fig. 5, there may be other causes of the cold-posterior effect, e.g. because inference is inaccurate (compensating for additional noise added in SGLD), or may compensate for issues arising from model misspecification (Grünwald, 2012; Grünwald et al., 2017). As such, we urge practitioners to regard tempering with caution: if a very large amount of tempering is necessary to achieve good performance, it may indicate issues with either inference or the prior, and fixing these issues is of the utmost importance to obtaining accurate uncertainty estimation. More importantly, we hope that our work will prompt more careful dataset design, and further study of how data curation might impact downstream analyses in machine learning. Indeed, there are initial suggestions that semi-supervised learning methods are also exploiting the artificial clustering (Fig. 3) induced by data curation (Aitchison, 2020).

ACKNOWLEDGEMENTS

I would like to thank Adrià Garriga-Alonso and Sebastian Ober for insightful discussions, Stoil Ganev for the GZ2 analyses which sadly came in only during the ICLR review period at which point author changes are banned (which is odd, given that revising the paper during the review period is allowed) and to Mike Walmsley and Sotiria Fotopoulou for getting us set up with GZ2. I would also like to thank Bristol's Advanced Computing Research Centre (ACRC) for providing invaluable compute infrastructure that was used for all the experiments in this paper.

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

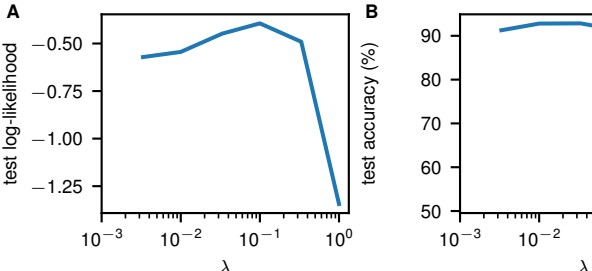

Figure 8: SGD in a ResNet18 with batchnorm removed with a tempered maximum a-posteriori loss. **A** Test log-likelihood. **B** Accuracy.

## A COLD POSTERIORS IN MAXIMUM A-POSTERIORI INFERENCE

Our explanation for the cold posterior effect suggests that it is not just due to the stochasticity of the posterior, but should also arise in e.g. maximum a-posteriori (MAP) inference. However, MAP inference causes serious issues in networks with batchnorm. In particular, with batchnorm, the outputs become invariant to the scale of the weights, so the weights can continually decay towards zero (Van Laarhoven, 2017). Note that these considerations are not relevant when we are doing full approximate inference, as the product of the prior and likelihood is a static quantity with a well-defined scale that cannot e.g. decay towards zero. As such, we considered tempering in our usual ResNet18, but where we have deleted the batchnorm layers. We used a standard protocol for these types of network: SGD with momentum of 0.9 and an initial learning rate of 0.1, followed by decays to 0.01 and 0.001 at epochs 150 and 200. We read off performance at the final epoch (250) and used a batch size of 128. We use a standard prior over the weights, with variance $2/\text{fan-in}$, to keep the scale of the inputs and outputs similar.

Remarkably, we found extremely strong cold-posterior effects (Fig. 8). These are perhaps expected as it is common-knowledge (but difficult to find a reference for) that while weight-decay is closely related to MAP inference with a Gaussian prior, good settings for the weight-decay coefficient are much smaller than those suggested by MAP inference with a sensible prior. Interestingly, these results suggest that the robustness of standard models to very low temperatures might arise due to batchnorm, and not be a fundamental property (e.g. having very large amounts of data).

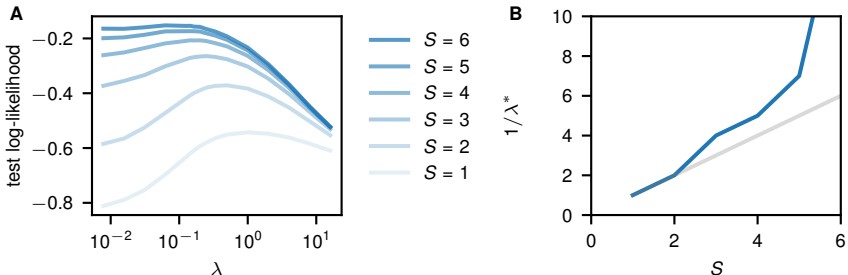

Figure 9: Toy BNN model. **A** Test log-likelihood vs $\lambda$ for different settings of $S$. **B** The optimal $\lambda$, plotted against $S$.

## B  TOY BAYESIAN NEURAL NETWORK EXAMPLE

To confirm the results in the Gaussian processes toy example in the main text, we did the same exercise with a toy Bayesian neural network. In particular, we considered 100 training examples, drawn IID from standard Gaussians, each with 5 input dimensions and 2 class outputs. We generated the "true" or target outputs from a one hidden layer ReLU network, with 30 hidden units, drawn from the prior. We trained the same network, using Langevin dynamics (full batch, but no rejection step; initialized randomly from the prior). As in the GP example, we applied our generative model of dataset curation with different settings for $S$ in the generative model. For test and training, we used the standard single-labeller log-likelihood, which ignores rejected noconsensus inputs. We implemented a highly parallelised Langevin sampling algorithm, which allowed us to draw 500 different datasets, and run one chain for each dataset (and each setting of $\lambda$ in parallel). The large number of chains allowed us to obtain very accurate results, but did mean that it was not possible to plot lines for individual datasets. We used a learning rate of 0.01 and burn in of 4,000 steps; to compute performance, we averaged over 50 samples of neural-network parameters, which were taken from a chain that was 20,000 steps long in total.

In Fig. 9A, we plot the test-log-likelihood for different settings of $\lambda$. There is a clear trend that as $S$ increases, the optimal temperature falls, and at the same time, the curves become flatter. Notably, these curves become flatter at the top for larger settings of $S$, making it difficult to identify the optimal setting for $\lambda$ at higher values of $S$. We then plotted the optimal setting of $\lambda$ against $S$ (Fig 9B). Initially, $S = 1$ is equivalent to the standard classification setup, as consensus is always reached, and as such, we find that the optimal temperature was 1. As in Fig. 4E, we found that $\lambda^* = 1/S$ held only approximately. Nonetheless, these results suggest that we might need more tempering than suggested by $\lambda^* = 1/S$, which supports our proposed explanation of the cold posterior effect. Finally, it should be noted that these results depend sensitively on the accuracy of the inference algorithm, so it is difficult to say as yet the degree to which mismatch between $\lambda^*$ and $1/S$ arises due to inaccuracy in inference or would emerge in the intractable true-posterior.

## C    GALAXY ZOO 2 DETAILS

The Galaxy Zoo 2 data set (Willett et al., 2013) was constructed by presenting images of galaxies to volunteers, who were asked to answer a questionnaire about the morphological features of the presented galaxy. These questions are not independent from each other but are part of a decision tree. As such, the questions a volunteer gets asked depends on their answers to earlier questions. For each galaxy, answers from multiple volunteers were collect. The raw data is recorded as counts of how many times each answer was picked across all volunteers.

From this data we wanted to derive classes in such a way that the classification performed by each volunteer was equivalent to a vote towards one of our classes. The most straightforward approach is to consider each unique path in the decision tree to be a unique class. Given that a volunteer fills in the questionnaire only once, their classification is equivalent to a single vote towards a specific class. The problem with this approach is that there are 1265 possible paths in the original decision tree. The number of volunteers per galaxy is less than a hundred, which means that the majority of these paths would receive 0 votes.

In order to address this, we simplified the tree to one where there are only 9 possible paths. The simplifications was performed by manually pruning branches of the decision tree. This way each path in the original tree could be mapped to a path in the simplified one, while maintaining specific aspects of its meaning. The simplified tree can be seen in (modified_graph.png). The following pruning operations were performed in order to create this simplified tree. Here we refer to each question using its task number, as given in the original paper.

- Task 02, Answer 'no': leads directly to Task 05, skipping Task 03, Task 04, Task 10 and Task 11
- Task 05, Answers 'dominant' and 'obvious': merged together becoming a combined category labelled 'obvious'
- Task 05, all answers: lead directly to the end of the graph, becoming exit points
- Task 07, all answers: lead directly to the end of the graph, becoming exit points
- Task 09, Answers 'rounded' and 'boxy': merged together becoming a combined category labelled 'with bulge'
- Task 09, all answers: lead directly to the end of the graph, becoming exit points
- As a consequence of the above pathing changes Task 03, Task 04, Task 06, Task 08, Task 10 and Task 11 are no longer reachable and thus removed from the graph

If we are to take each of the 9 paths in the simplified tree to form a class, then the resulting 9 classes would have the following meanings:

- smooth galaxy, completely round shape
- smooth galaxy, in-between round and cigar shaped
- smooth galaxy, cigar shaped
- edge-on disk galaxy, with central bulge
- edge-on disk galaxy, no central bulge
- face-on disk galaxy, obvious central bulge
- face-on disk galaxy, just noticeable central bulge
- face-on disk galaxy, no central bulge
- star or artifact

Using the simplified decision tree, we can decompose the data into votes for each path. The Galaxy Zoo data was recorded as counts for each specific answer, thus losing the information about which route a specific volunteer took in the tree. However, with the help of network flow algorithms it is possible to translate the data into counts of how many volunteers took a specific path. This way, by taking each path to be a class and the number of volunteers that followed a path to be the number of votes toward a class, we have a data set where each data point has received multiple classification over a fixed range of classes.

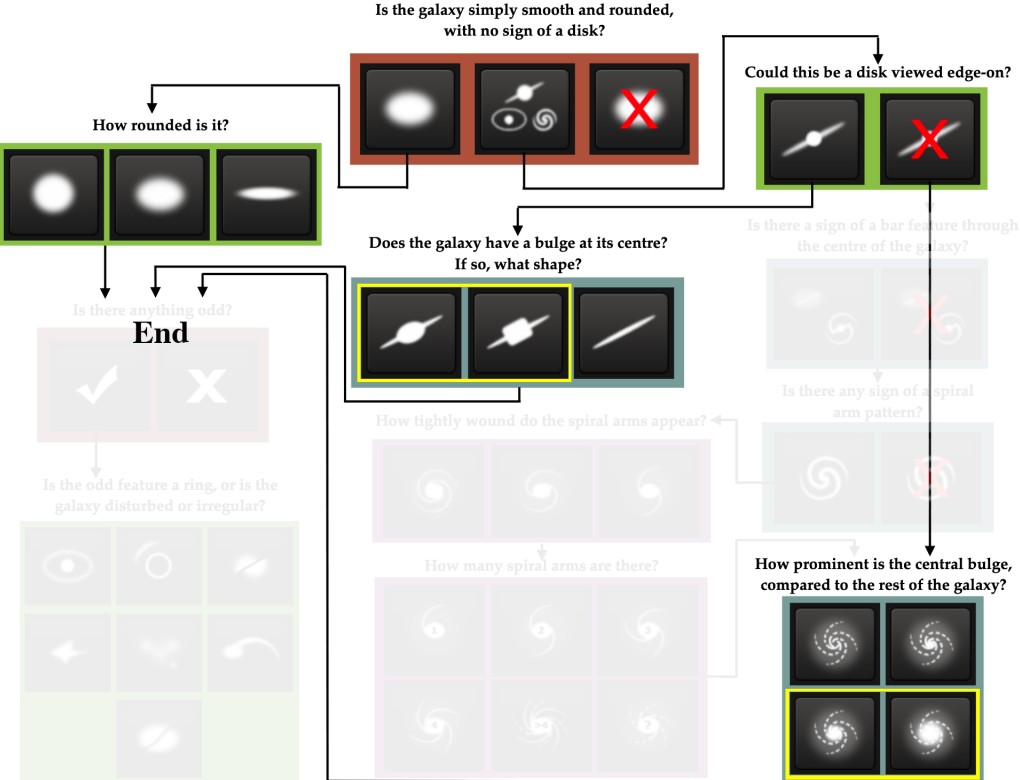

Figure 10: Modified decision tree from Willett et al. (2013) showing how we truncated the decision tree. Images in the same box are viewed as the same class.

For each galaxy in the Galaxy Zoo 2 data set we also have the image that was presented to the volunteers. As such, we can use our derived classes in an image classification task. In order to do this, we performed certain pre-processing. The original images have a resolution of 424x424 with the galaxy being at the center of the image. Thus, we performed a center crop of size 212x212 and then resized the images down to 32x32. For data augmentation, we used full 360 degree continuous rotations.

