# OpenReview forum: "A statistical theory of cold posteriors in deep neural networks"
_ICLR.cc/2021/Conference — ICLR 2021 Poster_

### Official Review · AnonReviewer1 · 2020-10-26
**Interesting theory, but the experiments are not quite there yet**

**Rating:** 6
**Confidence:** 4

**Review:**

The authors propose the idea that cold posteriors in Bayesian neural networks could be caused by the likelihood instead of the prior. They argue theoretically that the curation process of popular benchmark data sets would lead to a different weighting of the likelihood in the posterior. They show in some experiments that the cold posterior effect can be reduced when accounting for this.

Major comments:
- The paper title suggests that it is about cold posteriors, and it quite prominently references [1] in the introduction. However, in Sec. 2, it is then clarified that the paper is in fact not about cold posteriors, but about tempered ones. It is just briefly mentioned in passing that the results should transfer, but this is never tested. I think an experiment on actual cold posteriors, similar to the one in [1], would be warranted to support such a statement and the usage of the current paper title.
- The theory suggests that the optimal posterior performance should be achieved at lambda=S, that is, cooling down beyond that point should deteriorate performance again. This is an interesting prediction, since it does not seem to fit the observations in [1]. It would be nice to see this confirmed on an actual BNN experiment, similarly to what can be seen in the toy GP experiment.
- The related work section seems awfully short. It does not even mention [1] (although it is cited heavily elsewhere). Moreover, for a paper that is proposing a statistical theory of tempered posteriors, works such as [2] and [3] should probably be mentioned.

Minor comments:
- Sec. 4.2: "we when" -> "when"

Summary:
The idea that cold/tempered posterior effects can be caused by the data set curation instead of by misspecified priors is very interesting and definitely deserves a theoretical and empirical investigation. However, the investigation at hand seems a bit incomplete in some places, especially the related work and experiments sections. Also the title does not currently seem to fit the experiments. Given that the current manuscript is comfortably within the ICLR page limit, I'm hopeful that these points can be addressed in a revised version during the discussion phase.

Update: I increased my score following the clarifications and addition of the BNN experiment during the discussion phase.

[1] Wenzel, F., Roth, K., Veeling, B. S., Świątkowski, J., Tran, L., Mandt, S., ... & Nowozin, S. (2020). How good is the bayes posterior in deep neural networks really?. arXiv preprint arXiv:2002.02405.
[2] Grünwald, P. (2012, October). The safe bayesian. In International Conference on Algorithmic Learning Theory (pp. 169-183). Springer, Berlin, Heidelberg.
[3] Grünwald, P., & Van Ommen, T. (2017). Inconsistency of Bayesian inference for misspecified linear models, and a proposal for repairing it. Bayesian Analysis, 12(4), 1069-1103.

---

> ### Author Response · Authors · 2020-11-18
> **Response**
>
> We have added an additional experiment in which the cold posterior effect disappears as we add noise to CIFAR-10 labels.  This is expected under our theory, as one of the key implications of dataset curation is that the labels are very likely to reflect the true classes.
>
> * We have updated Sec. 2 to describe AnonReviewer4's point that the only difference between cold and tempered posterior is whether we apply scale the prior variance. So cold and tempered posteriors do not differ (at least in the standard setting of Gaussian priors), and our results relate to both tempered and cold posteriors.
>
> * Wenzel et al. 2020 did find that performance deterioriates as you continue to reduce the temperature (their Fig. 5+6, for the cross-entropy for the largest batch sizes).  Note that in SGLD, larger batches give more accurate posterior sampling.  For smaller batches, the noise due to minibatched gradient estimation is larger, and it may be that smaller temperatures serve to partially compensate for this additional noise.  That said, even under our theory of dataset curation, we do not necessarily expect large drops in performance as temperature continues to increase.  If there is alot of data, then both the Bayesian and maximum likelihood solutions will be very close to the optimal settings of the parameters.  In that case, sending $\lambda\rightarrow 0$, which takes us from the Bayesian posterior to the maximum-likelihood solution, is unlikely to make much difference to the inferred parameters or predictive performance.  We are aiming to complete the toy BNN experiments by the end of the review period (we wanted to get the initial responses up to allow for discussion).
>
> * We have expanded the related work section including your references [1-3].  "Wenzel et al. (2020) introduced the cold-posterior effect in the context of neural networks, and proposed and dismissed multiple potential explanations (although none like the one we propose here). Other past work, though not in the neural network context, argues that tempering may be important in the context of model misspecification (Grünwald, 2012; Grünwald et al., 2017).  Critically, we believe that there may be many causes of cold-posterior like effects, including curation, model misspecification and artifacts from SGLD.  Ultimately, the contribution of each of these factors in any given setting will depend on the exact dataset, model and inference method in question.  Importantly, this also means we do not necessarily expect there to be no tempering in uncurated data, our theory demands merely that we see _less_ tempering in the case of uncurated data."
>
> Minor comments:
>   Fixed.

---

> > ### Comment · AnonReviewer1 · 2020-11-18
> > **Thanks**
> >
> > Thanks for the clarifications and the additions to the related work section. I'm looking forward to seeing the results of the BNN experiment. Should they fit your theoretical predictions, I would be happy to increase my score.

---

> > > ### Author Response · Authors · 2020-11-19
> > > **BNN experiment is up!**
> > >
> > > Thanks again for your comments!
> > >
> > > The BNN toy example is now in Appendix B.  We see the same general pattern of $\lambda^* \approx 1/S$ holding approximately.  If anything, it appears that $1/\lambda^*>S$, which implies $\lambda^* < 1/S$, i.e. we need _more_ tempering than that suggested by $S$ alone.
> > >
> > > Please let me know if you have any further questions/would like any further clarifications.

---

> > > > ### Comment · AnonReviewer1 · 2020-11-19
> > > > **Nice results**
> > > >
> > > > Thanks for adding the experiment, the results look nice. I increased my score.

---

### Official Review · AnonReviewer2 · 2020-10-28
**Interesting theory but lacking evidence.**

**Rating:** 6
**Confidence:** 4

**Review:**

The work propose a theory suggesting that the cold posterior phenomena arises solely due the the curated nature of image benchmarks. A generative model is proposed where multiple annotators label datapoints, and only unanimously labeled datapoints are accepted into a dataset. This theory is studied under a toy-problem using VI and a relabelled version of the CIFAR-10 test set with SGLD.

However, many questions remain unanswered and the proposed theory is not sufficiently studied.

- Q: The cold posterior problem was highlighted in the SGMCMC case, but this work's main toy problem only explores tempered posteriors as prevalent in VI. It would be beneficial if the work highlights why these results should extend to the cold posterior, or better yet, run experiments in this scenario.
- Q: 4.1 strongly suggests that there is a relationship between between \lambda and S in the toy problem. It should be an easy addition to study this connection for a range of values for S to explore if this holds.
- Q: The work claims that the consensus protocol for standard datasets is not available, but it would appear that this is a a simple manner of reaching out to the authors of the datasets.
- Q: The main experiment presented in figure 5 is missing some important ablations: what happens when the CIFAR-10 baseline is trained under the same conditions (learning rate) as CIFAR-10H?
- Q: Why is it acceptable to use the original CIFAR-10 training-set as the test-set for CIFAR-10H? This seems like a problematic shift in data distribution.
- Q: the theory of dataset curation is interesting but makes a broad claim. More datasets should be explored from varying modalities. The sole focus here on CIFAR-10 provides too little evidence. As a suggestion: curation processes are different for e.g. medical imaging datasets, and typically well documented.
- Q: It's unclear to me why it is acceptable to increase the training set size by the number of annotators. An increase of factor 50 is effectively setting the temperature to ~ 1e-2. At this temperature the baseline performs just as well. Does this large gap still hold if just a much smaller subset of annotators is taken from the CIFAR-10H dataset?

The core idea proposed in this work is thought provoking and contributes to the discussion on this topic. The work is relatively short, which is not a problem in its own right, but the experiment section needs to provide more evidence and analysis. I vote for reject.

Nitpicks:
- F instead of E in figure 4.
- "As expected, we when"

Update: I've increased my score.

---

> ### Author Response · Authors · 2020-11-18
> **Response (1/2)**
>
> Thanks for your comments!
>
> We have added an additional experiment where we break the statistical properties introduced by consensus by adding noise to CIFAR-10 labels (new Fig 6).  Adding label noise eliminates and eventually reverse cold-posterior effects.
>
> * We have updated Sec. 2 to point incorporate AnonReviewer4's point that the only difference between cold and tempered posterior is whether we apply temperature scaling to the prior.  In the case of Gaussian priors, this rescaling can be absorbed into the prior variance.  As such, any tempered posterior is equivalent to a cold posterior under a rescaled prior.
>
> * We have updated Fig. 4 to examine this relationship in-depth.  We find that $\lambda^* = 1/S$ holds approximately, but probably not exactly. This is expected: we have added a new Sec. 4 discussing the differences between the data curation and cold/tempered-posterior setups, which imply that we cannot expect $\lambda^*=1/S$ to hold exactly.
>
> * The consensus protocol is usually described in the publications introducing the datasets, and our consensus protocol is a simplified version of these.  What isn't usually available is the images they considered and discarded as being ambiguous. In the case of CIFAR-10 in particular, they considered a subset of images from the 80 million tiny images dataset, and this dataset has been formally withdrawn ("https://groups.csail.mit.edu/vision/TinyImages/") [1], due to ethical issues.
>
> * We are training with same _effective_ SGD learning rate.  Consider a CIFAR-10H image with 50 labels, all the same.  In the Bayesian setting, we have to condition on all 50 labels.  So the loss, and hence the gradients are 50 times larger.  As such, the effective learning rate (i.e. the size of a single gradient step) is 50 times larger, and this breaks the SGLD algorithm (you can't set the effective learning rate 50 times larger than the standard value and expect everything to keep working).  To keep the effective learning rate (i.e. the size of a single gradient step) the same, we decrease the actual learning rate by a factor of 50.  An alternative way to think about this is that it is equivalent to using the average log-likelihood over labellers (rather than images) as the loss.  We have updated the text to reflect this.  Note that all other parameters (number of cycles, length of cycles, length of training set etc.) are exactly the same.  Moreover, in the CIFAR-10 comparison, we use the CIFAR-10 test set for training, just as we do for CIFAR-10H.  Given that the protocols are exactly the same, we expect performance to become very close as $\lambda\rightarrow 0$, as in this limit both methods ignore the prior and revert to maximum likelihood.  This convergence is indeed observed (Fig. 5).
>
> * The input images for CIFAR-10H are exactly the CIFAR-10 test set.  As such, input images for CIFAR-10H and the CIFAR-10 training set are drawn IID from the same distribution.  Thus, the only place that differences could arise is in the labels.  Remember that we interpret the $\sim 50$ CIFAR-10H labels, $Y_s$, and the CIFAR-10 labels, $Y$, as observations at different points within the same graphical model (Fig 2B or C). As such, what really matters is whether the CIFAR-10 $Y$'s and the implied CIFAR-10H $Y$'s are IID.  We used a majority vote as the as the proxy for the CIFAR-10H $Y$, and found that in 99.21% of cases, this majority vote matched the CIFAR-10 label.  Such a difference could arise through stochasticity even in the case of IID labels, but it does indicate that any difference in the distribution of the CIFAR-10 and CIFAR-10H label distributions must be minimal.
>
> * We hope that our work will raise awareness of the potential issues regarding data curation, and prompt a number of careful studies of its effects. For the moment, many of the studies on e.g.\ cold posteriors were developed on CIFAR-10 (which is presumably an exemplary case of this type of curation), so we believe that it is justified to focus on CIFAR-10 in this initial work.  In general, we hope that medical imaging datasets will be *less* curated and hence less vulnerable to these effects, as ground-truth can often determined e.g. by a biopsy, rather than just by looking at the image, in which case there is far less justification for throwing away images.
>
> [1] Prabhu VU, Birhane A. Large image datasets: A pyrrhic win for computer vision?. arXiv:2006.16923. (2020)

---

> > ### Author Response · Authors · 2020-11-18
> > **Response (2/2)**
> >
> > * As we are in a Bayesian setting, we have to condition on all the data.  If we have 50 labels for one data point, the data is those 50 labels and we have to condition on all of them.  We have no choice.  As you say, this _in effect_ sets the temperature to $\sim 10^{-2}$ (relative to just having a single label).  And this is the key observation behind our approach.  Our contention is that even though the CIFAR-10 dataset gave us one label, during the curation process, there were multiple labels by multiple individuals, all in agreement, and because we are being Bayesian we _need_ to take the true generative process, which did include those additional labels, into account.
> >
> > Nitpicks:
> >   * Fig. 4 has been reworked, but has correct panel labels.
> >   * Fixed

---

> > > ### Comment · AnonReviewer2 · 2020-11-20
> > > **Response to last bullet.**
> > >
> > > * I see your point, and I think this is a very interesting discussion! I would argue that there is a law of diminishing returns of additional annotators  that does not warrant the linear inverse proportional rule. If 10 annotators agree on the same label for a datapoint, why would having an additional 40 annotators agree on the same label warrant an additional 1/5 reduction of the temperature? I'm also not sure that it's fair to say that 50 duplicated labels can justify the same temperature decrease as could 50 times as many datapoints. The later surely provides more posterior evidence from a bayesian perspective.  What justifies setting the number of annotators to exactly 50? Why not 5, or 500?
> > >
> > > The problem with this approach is that you could use it to justify any temperature reduction. If a model is optimal at t=0.0001, we just claim that there were 10.000 annotators.

---

> > > > ### Author Response · Authors · 2020-11-21
> > > > **Response**
> > > >
> > > > Thanks for your comments again!  There are two points to make here:
> > > > 1. In the case of CIFAR-10H, there really are ~50 labels (and they aren't all duplicated), meaning for Bayesian inference, we really do have to take all of them into account --- again the data is the data, and we can't downweight it based only on intuitive arguments.
> > > > 2. That said, you raise an important point about the practical application of our theory: namely that it is difficult to know the "right" value for S in practical settings such as CIFAR-10, where our generative process was not followed   exactly.  In particular (as quoted in the paper) instructions such as "It’s worse to include one that shouldn’t be included than to exclude one" (Krizhevsky 2009; introducing CIFAR-10) are likely have the effect of increasing the effective value of S, but it isn't clear how much.  So in practical settings, all we can do is find the optimal value of S.  Importantly, this inability to determine the "true" value of S merely means that we can't test the theory in this way (it this doesn't give evidence against the theory).  But we agree there is a risk that downstream practitioners may use our arguments to "to justify any temperature reduction".  As such, we have modified the Conclusion section to "Discussion and Conclusion" in which we explicitly discuss this risk, and argue that tempering should be regarded with extreme caution, as it can also arise from model misspecification or inaccurate inference.

---

> > ### Comment · AnonReviewer2 · 2020-11-20
> > **Thoughts**
> >
> > Thank you for addressing my concerns!  Please see below.
> > - Agreed!
> > - Agreed!
> > - Re: consensus protocol. On my reading of the manuscript it did not stood out to me that you use a simplified version of the official consensus protocol, but perhaps I missed this.
> > - Re: 'We are training with same effective SGD learning rate'. Thanks for the clarification! Am I correct to understand that this implies that the gradients of the loss are then equivalent, and that - in this experiment - the only difference between these two methods is essentially the reduced temperature. Or equivalent, a shift in the x-axis? Not to say that theres anything wrong with that, just trying to understand how to interpret this plot. If not, are there any remaining ablations that can be made to explore where the different behaviour comes from? If so, I think this should be clarified to the reader, so that the results can be correctly interpreted especially regarding the point in my other comment.
> >
> > - Regarding datasets, I think the point remains that a claim about data curation should be explored in more than one datasets and modality to ensure that the insights generalize. In the case of medical image analysis, especially in segmentation, different annotators have highly varying boundaries and the consensus protocol is non-trivial (see e.g. the analysis in the probabilistic unet paper https://arxiv.org/abs/1806.05034). That said, I understand that extending the bayesian treatment to segmentation is a complicated endeavour and too much to ask for in an initial work.
> >
> > Overall, I remain somewhat skeptical that the curation process is the sole reason for the cold posterior phenomena and I think the effect is overstated. However, I do agree that it could be a piece of the puzzle, and it is certainly a novel and cool insight. I think the paper would improve if the limitations of this theory are explored as well, but I will increase my score as it already provides interesting insights in its current form.

---

> > > ### Author Response · Authors · 2020-11-21
> > > **Response (with new dataset!)**
> > >
> > > * .
> > > * .
> > > * This is described at the start of Sec. 3.  Let us know if you'd like any clarifications to that Section!
> > > * The gradients of the log-likelihood point in almost, but not quite the same direction, because there are some disagreements in the CIFAR-10H dataset.  But we have added a note about the "trivial" interpretation of this experiment.
> > > * We have added a new dataset based on Galaxy Zoo 2 (GZ2).  This is a super-interesting dataset for our purposes, because it is not curated (the celestial objects to image were chosen merely based on e.g. brightness and spatial extent), and because the classifications of ~50 labellers per image are available.  As such, we were able to use GZ2 to define a closely matched curated and uncurated datasets. We defined an uncurated dataset by choosing images at random, and we  defined a curated dataset by choosing the most confident images in each class such that we achieved the required dataset size while maintaining class balance.  (Note that we were able to directly use our generative model of curation, because it massively changed the number of each class in the resulting dataset.)  As expected, we found that the curated dataset exhibited large cold-posterior effects, while the uncurated dataset did not. We hope that these additional experiments demonstrate significant cold-posterior effects can arise due to curation in real-world datasets.
> > >
> > > Finally, we absolutely agree that curation is not the only cause of cold-posterior effects, and have updated the manuscript to emphasise that. We added "Critically, we believe that there may be many causes of cold-posterior like effects, including curation, model misspecification and artifacts from SGLD. Ultimately, the contribution of each of these factors in any given setting will depend on the exact dataset, model and inference method in question. Importantly, this also means we do not necessarily expect there to be no tempering in uncurated data, our theory demands merely that we see less tempering in the case of uncurated data." to related work in response to AnonReviewer1, and we added "As such, we urge practitioners to regard tempering with caution: if a very large amount of tempering is necessary toachieve good performance, it may indicate issues with either inference or the prior, and fixing these issues is of the utmost importance to obtaining accurate uncertainty estimation" in response to your other comments.

---

### Official Review · AnonReviewer3 · 2020-10-29
**Sensible idea and very well executed, convincing results, but some questions remain.**

**Rating:** 7
**Confidence:** 3

**Review:**

This paper addresses the perplexing issue of cold posterior having better predictive performance than the ideal Bayesian posterior in Bayesian deep learning (Wenzel et al., 2020), and offers a possible explanation in terms of a mis-specified likelihood function that deviates from the true generative process of the data. By considering the data curation process and augmenting the likelihood model accordingly, the effect of cold posterior is shown to diminish significantly, and the ideal posterior is again optimal. Empirical results on both a toy problem and image classification support the theory.


------------------

Pros:
1. Given the prevalence of Bayesian deep learning and the issue of cold posterior, this paper offers a timely contribution that bridges theory and practice.
1. The paper is well written and motivated, the method appears sound (but see questions below), and concepts are explained in a clear and pedagogical manner.
2. The experiments are well thought out and offer clear empirical support of the proposed hypothesis.

------------------

Cons:
This might be due to my limited understanding of the paper, but I think there are still some limitations to the paper's proposed theory, e.g., it doesn't explain the observation that extremely cold posterior (λ -> 0) doesn't seem to hurt the performance of BNN (which should, according to the proposed theory, as there is only one optimal temperature λ = 1 / S, where S is the true number of underlying labelers), and more below.

------------------

Questions and Comments:
1. My biggest confusion is this: the paper argues that it's incorrect to assume a simple categorical likelihood p(y|x) as it doesn't take into account the data curation process; however, under the extended likelihood model as proposed, when conditioning on the event that y!=None and x!=None (as we do when training on standard datasets), and after marginalizing out the intermediate variables and renormalizing, isn't the conditional distribution p(y|x) still just a categorical distribution (except parameterized in a different way now)?  If so, then the difference between the two likelihoods is really just a different parameterization, and I'm no longer sure what to make of the suggested theory and the supporting results.  I find it very surprising that the more complex parameterization significantly reduced the tempering effect.  And if the we take the ground truth likelihood p(y|x) to be as in the standard (curated) dataset, which the paper argues is in some sense artificially "tempered", then why can't a well-spcified BNN just adapt to this (still categorical) likelihood and learn an optimal posterior under it?  I'm happy to raise my score if the authors can clarify these issues for me.
2. Since point estimation with SGD optimizes the same likelihood function, why don't we observe the tempering effect in SGD? Perhaps there is some effect but rather minimal; in any case, some experiments on SGD (with / without the corrected likelihood) would be interesting.
3. Related to my comment about "extremely cold posterior": in the GP experiment, when trained and tested on the corrected likelihood (considering curation), the test performance seems to really prefer the optimal λ = 1, whereas on the image experiment, more tempering (λ -> 0) doesn't seem to affect test performance. Is there an explanation for this?
4. Finally, does the proposed theory explain the observation that the cold posterior effect is more prominent in BNN with higher capacity (Wenzel et al. 2020)?


------------------

Possible typos and minor mistakes:
1. p.3, under eq (7), " This likelihood is equivalent to labeling each datapoint S times with the same label, and therefore has exactly the effect of setting λ = S in a tempered posterior". Should be "λ = 1/S" instead.
2. The right-most subfigure in Figure 4 should be labeled "E" instead of "F" to match the caption below.


------------------
Update:

I've raised my score in light of author response and new results.

---

> ### Author Response · Authors · 2020-11-18
> **Response**
>
> Thanks for your comments!
>
> 1. Great comment.  We have rewritten and hopefully clarified Methods (Sec. 3) to clarify this point.  We begin by considering the input image known, but the label might be None.  In that case, there are two probabilites we could consider: not conditioning on consensus, $P(Y| X, \theta)$ (Eq. 6) or conditioning on consensus, $P(y| X, \theta, y\neq \texttt{None})$ (Eq. 10). As you suggest, the conditional, $P(y| X, \theta, y\neq \texttt{None})$ (Eq. 10), just represents a reparameterisation of the softmax. But $P(Y| X, \theta)$ (Eq. 6) does not (see Eq. 8 and 9).  This raises the question: should we use Eq. (6) or Eq. (10) for Bayesian inference in the standard setting where the no-consensus data has been thrown away?  To answer this question, we need to carefully write down the full joint distribution (Fig. 3C), and marginalise over the unknown latents.  At the end of these derivations (Eq. 15), we end up with an expression that is analogous to use Eq. (6) (i.e.\ the one that doesn't condition on consensus, and isn't equivalent to reparameterisation of the Categorical softmax).
>
> 2. We do!  Specifically, it is well known that setting a Gaussian prior on the weights and doing MAP inference is equivalent to doing standard maximum-likelihood with weight decay.  Critically though, Gaussian priors on the weights (with sensible variances) almost always suggest weight-decay coefficients that are far larger than the empirical optimum. These artifically smaller weight decay coefficients can be interpreted as tempering (i.e. increasing the importance of the likelihood relative to the prior).  We have run experiments (Appendix A, Fig 7) demonstrating these effects, which turn out to be extremely strong.
>
> 3. (Also Cons) If there is alot of data, then both the Bayesian and maximum likelihood solutions will be very close to the optimal setting of the parameters.  In that case, sending $\lambda\rightarrow 0$, which takes us from the Bayesian posterior to the maximum-likelihood solution, will make little difference to the inferred parameters or predictive performance.
>
> 4. We suspect that for small (low capacity) networks, relatively little data is needed to pin down the parameters close to their optimal values.  Once the parameters are close to optimal, more data or more tempering will not give any improvement in predictive performance.  The parameters of larger network are likely to be further from the optimal parameters (and perhaps closer to the prior), giving more room for improvement when we introduce tempering.
>
> Possible typos and minor mistakes:
>   * Fixed
>   * Fig. 4 has been reworked, but has correct panel labels.

---

> > ### Comment · AnonReviewer3 · 2020-11-25
> > **Thanks for the response and updated results. Some clarification on the role of the prior v.s. likelihood in tempering would be helpful.**
> >
> > Thank you for your thorough response and clarifying my doubts. I've raised my score in light of the author response and new results. Response to your response:
> >
> > 1. That makes sense. What I was describing earlier would require a different graphical model than proposed in Fig. 2C, where a newly introduced binary variable $C$ would indicate consensus (which we treat as side data, like $X$), $p(Y|\theta, X, C=\text{True})$ is the "reparameterized" categorical distribution (Eq.10), and there would be no arrow going from $\theta$ to $C$ (i.e., the consensus process doesn't depend on $\theta$), so that the MAP objective is $\log p(\theta | X, C, Y) = \log p(\theta) + \log p(Y|\theta, X, C) + \text{const}$, corresponding to still a categorical likelihood. But yeah, in your proposed model, the likelihood for Bayesian inference is no longer categorical. As a comment for future work: since we are now basically asking "which generative model is the correct one for the data?", perhaps Bayesian model comparison could offer an alternative and more systematic way to answer this, than examining if a model experiences cold posterior effect.
> >
> > 2. Good to know that the tempering effect also exists in MAP inference.
> > (a). However, can this tempering effect be caused by too strong of a prior? The experiment used "Gaussian priors on the weights (with sensible variances)", but how do we know a priori if the prior variances are sensible?  More generally, this relates to my other confusion with this paper, that is, the proposed curation model indeed seems to explain the tempered posterior effect, but how do we know it's the likelihood but not the prior at fault? In MAP inference, up-weighting the log-likelihood term is equivalent to down-weighting the log prior term, so it's possible the "sensible" prior variances we start with are too small, and the improved performance from tempering is due to broadened prior (this complementary effect of prior v.s. likelihood no longer holds exactly in the full Bayesian inference setting, but I expect something similar is happening). If this were the case, then it seems to contradict the finding of Wenzel et al. (2020), whose prior variance scaling experiment (see their Fig 12) didn't seem to alleviate cold posterior effect.
> > (b). There might be an obvious reason for this, but I'm curious why $\lambda=1$ gave poor test accuracy in Fig 8 B., yet the highest test log likelihood in Fig 8 A.
> >
> > 3 & 4. Thanks for your explanation.
> >
> >
> >
> > Additional comments and suggestions:
> > 1. For the experiments (e.g., in Fig 4.C and Sec 5.3) that trained/inferred with the correct likelihood, i.e., Eqs (6) and (7), it's somewhat unclear how the test log likelihood is calculated. Did it still use Eqs (6) and (7), or the categorical probabiliy Eq (10) as suggested in section 4? Which one is more correct and does it make much difference?
> > 2. It can be helpful to have a table describing high-level setups and findings of the extensive experiments, particularly how the data is generated (curation v.s. no curation), what form of likelihood is used for inference/training (equation numbers), and similarly how the test log likelihood is computed (which can benefit from explicit definitions).
> > 3. Nitpicks:
> > (a). Eq(15): X should be replaced with Z, since Eq (12) integrates X out with respect to the Dirac measure \delta(Z-X).
> > (b). Sec 5.3: "Importantly, this change in learning rate, leaves the stationary distribution was unchanged" has an extra "was" in the sentence.

---

### Official Review · AnonReviewer4 · 2020-11-03
**What a beautiful paper!**

**Rating:** 9
**Confidence:** 4

**Review:**

Apologies for the late review!

## Summary of the Paper

The paper provides a potential theoretical explanation of the known empirical observation that cold (or tempered) posteriors improve predictive performance of deep Bayesian neural networks. The provided explanation is simple and it leads to additional predictions, which the authors check empirically as far as possible with existing data sets. The empirical results agree with the predictions.

## Main Strengths

I believe the main message of the paper is so relevant and seems so simple (at least in hindsight) that it has the potential of becoming a kind of "common knowledge" in Bayesian neural networks community (caveat: I can't judge if these findings had already been informally known to a larger group of researchers, but unless someone has explicitly written them down somewhere I wouldn't hold this against the paper).

Researchers have been trying to increase predictive performance of deep neural networks by applying scalable Bayesian methods to deep learning for a while, but even replicating the performance of point estimated models with Bayesian neural networks has proven surprisingly difficult. To my knowledge, it was found out only recently that Bayesian neural networks systematically outperform their point estimated counterparts if the prior is made artificially sharper than what probability theory would predict (i.e., by "lowering the temperature"). This empirical result has been puzzling from a theoretical perspective, but the present paper provides a simple potential explanation for this effect.

I believe the findings in this paper go beyond a theoretical justification of an empirically known fact. The findings may also have implications on model robustness: the authors argue that the "tempering effect" is a result of curation of the training set. Validations and tests sets are typically curated in the same way as the training set in the machine learning community. However, when models are deployed in the field, they typically see uncurated data points. I would be curious to know if explicitly modeling the curation process, as the authors do in this paper, would also address this issue.

## Potential Weaknesses

There's one caveat to my review: I am not an expert on Bayesian neural networks and, as stated above, the argument made in this paper seems so simple in hindsight that I cannot say with absolute certainty that it hasn't been made before. I personally haven't heard this argument before, but if other reviewers can point to a reference that already made this argument, then that would probably be the only thing that could convince me to lower my rating. Otherwise, I would consider the simplicity of the authors' argument a strength of the paper.

## Questions to the Authors

- What do the authors mean with the phrase "finite networks" in the first paragraph? Is it the same as networks with point estimated parameters (as opposed to Bayesian neural networks)?
- As mentioned above, the paper models the curation of the training set, but I didn't understand how or if curation of the test set is modeled. Could the authors clarify this? Specifically, what changes for a model trained on curated data when it is either (a) tested on an equally curated test set or (b) applied to uncurated data in the wild? Would the optimal $\lambda$ during training differ between cases (a) and (b) or would the posterior have to be changed after training?

## Minor Issues

- I like the short Section 2 (which compares "cold" and "tempered" posteriors) very much! I think it could even be improved by adding one reference each for "cold" and "tempered" posteriors, respectively. More importantly, as far as I understand, the two are really essentially the same if one uses, e.g., a Gaussian prior. Unless I'm mistaken, the missing factor of $\frac{1}{T}$ in Eq. 2 could then be absorbed into a rescaling of the prior covariance (unless the prior covariance itself is learned with expectation maximization). If this is correct then I'd add a corresponding statement at the end of Section 2 (it would make the paper's claims more widely applicable).
- I think Figure 2 is never discussed in the paper (but there should be enough space left to discuss it). The figure caption says "Schematic diagram". Could the authors clarify what this means? Do the points come from some toy model with a 2-d parameter space (or does the figure show a 2-d PCA of the parameter space) or were the points really just drawn manually to visualize the idea? I think both would be fine but I would be very curious to know how the figure looks with real data.
- In Figure 4, the last panel is labelled "F" but referred to as "E". Also, in the last two panels, the left dashed vertical bar is not discussed. Is it the theoretically expected optimal value of $\lambda$ (i.e., $\lambda=\frac{1}{4}$) or the empirically found optimal value?

---

> ### Author Response · Authors · 2020-11-18
> **Response**
>
> Thanks for your comments!
>
> * "finite" was left over from an old draft.  We have replaced it with "non-Bayesian".
> * Great question! We have added a brief discussion of this point in the new Sec. 4, which more broadly discusses differences between the cold/tempered-posterior and our model of dataset curation.  As usual in the Bayesian setting the posterior over the weights (and the optimal $\lambda$) depends only on training data, not test data.  That said, the predictive distributions differ depending on whether or not the test-set has been curated.  If the test set is curated, then the predictive distribution becomes the new Eq. 10, which in effect "tempers" the predictive distributions.  In contrast if the test-set is drawn from the "wild", not curated, and labelled by a single labeller, then we should use the single-labeller predictive distribution, $P(Y_s| X, \theta)$.
>
> Note that the test set for standard benchmark datasets is usually curated, but the cold-posterior setups use the single-labeller predictive, so this represents a difference between the cold-posterior and curated-data setups, which we discuss in the new Sec. 4.  The conclusion of this section is that while curation does provide an explanation of the cold posterior effect, differences in the setups mean that we can't expect exact relations like $\lambda^*=1/S$ to hold.  The exact relation is really an empirical matter, which is addressed in the toy data (expanded Fig. 4).
>
> Minor issues:
> * References included.  And great point! We've added a discussion of this point in Sec. 2.
> * We have added a discussion of this Figure in the main text, and added a more detailed description of its generation to the Figure caption: "The input points for "cat" and "dog" points are generated from separate 2D Gaussian distributions, and the classifier (and decision boundary) comes from the ratio of the Gaussian probability density functions.  For the consensus processes, we used $S=7$ (we used a relatively large value to make the effects unambiguous)."
> * Fig. 4 has been reworked, but has correctly ordered panel labels.  $\lambda=1/4$ is "expected" (due to the form for the likelihoods matching those for tempered data), but not guaranteed due to differences between the cold-posterior setup and our model of dataset curation (see the new Sec. 4).  Empirically, Fig. 4E suggests that $\lambda^*=1/S$ holds approximately, but perhaps not exactly.

---

### Decision · Program_Chairs · 2021-01-07
**Final Decision**

**Decision:**

Accept (Poster)

**Comment:**

This is an interesting, controversial paper that contributes to an ongoing debate in Bayesian deep learning.

Bayesian inference with artificially “cooled” posteriors (e.g., trained with Langevin dynamics with down-weighted noise) was recently found to outperform over both point estimation and fully-Bayesian treatments (Wenzel et al., 2020). This paper proposes a new explanation for these observed phenomena in terms of a data curation mechanism that popular benchmark data sets such as CIFAR underwent. The analysis boils down to an evidence overcounting/undercounting argument and takes into account that curated data sets only contain data points for which all labelers agreed on a label. The authors claim that, when modeling the true generative process of the data, the cold posterior effect (partially) vanishes.

The paper is well-written and provides a consistent analysis by modeling the data curation mechanism in terms of an underlying probabilistic graphical model of the labeling mechanism. Unfortunately, several observed phenomena of (Wenzel et al., 2020) remain unexplained by the theoretical arguments, e.g., the fact that “very cold” (T --> 0) posteriors don’t hurt performance, or the observation that the optimal temperature seems to depend on the model capacity. While the proposed explanation doesn’t capture the full picture (upon which both authors and reviewers agree), the paper’s focus on the data curation process, supported extensive experiments, gives a partial explanation and provides an interesting perspective that will spur further discussion and should be of broad interest to the Bayesian deep learning community.